# Impact of Rehabilitation on Physical and Neuropsychological Health of Patients Who Acquired COVID-19 in the Workplace

**DOI:** 10.3390/ijerph20021468

**Published:** 2023-01-13

**Authors:** Katrin Müller, Iris Poppele, Marcel Ottiger, Katharina Zwingmann, Ivo Berger, Andreas Thomas, Alois Wastlhuber, Franziska Ortwein, Anna-Lena Schultz, Anna Weghofer, Eva Wilhelm, Rainer-Christian Weber, Sylvia Meder, Michael Stegbauer, Torsten Schlesinger

**Affiliations:** 1Institute of Human Movement Science and Health, Faculty of Behavioral and Social Sciences, Chemnitz University of Technology, 09107 Chemnitz, Germany; 2BG Hospital for Occupational Disease Bad Reichenhall, 83435 Bad Reichenhall, Germany

**Keywords:** work-related COVID-19, physical capacity, neuropsychological health, work ability, rehabilitation

## Abstract

Workers, especially healthcare workers, are exposed to an increased risk for SARS-CoV-2 infection. However, less is known about the impact of rehabilitation on health outcomes associated with post-COVID. This longitudinal observational study examined the changes in physical and neuropsychological health and work ability after inpatient rehabilitation of 127 patients (97 females/30 males; age 21–69 years; Mean = 50.62) who acquired COVID-19 in the workplace. Post-COVID symptoms, functional status, physical performance, neuropsychological health, employment, and work ability were assessed before and after rehabilitation. Group differences relating to sex, professions, and acute COVID status were also analyzed. Except for fatigue, the prevalence of all post-COVID symptoms decreased after rehabilitation. Significant improvements in physical performance and neuropsychological health outcomes were determined. Moreover, healthcare workers showed a significantly greater reduction in depressive symptoms compared to non-healthcare workers. Nevertheless, participants reported poor work ability, and 72.5% of them were still unable to work after discharge from rehabilitation. As most participants were still suffering from the impact of COVID-19 at rehabilitation discharge, ongoing strategies in aftercare are necessary to improve their work ability. Further investigations of this study population at 6 and 12 months after rehabilitation should examine the further course of post-COVID regarding health and work ability status.

## 1. Introduction

Scientific insights about the prevention and treatment of acute SARS-CoV-2 infections have increased in recent months. Current results of reviews and meta-analyses have accumulated in international guidelines, e.g., in the World Health Organization (WHO, Geneva, Switzerland) and in national guidelines, e.g., of the National Institute of Health (Bethesda, MD, USA) or the German national S3-guideline [1].

The known transmission pathways of SARS-CoV-2 reflect that workplaces are generally a high-risk setting for the viral transmission of SARS-CoV-2 due to interpersonal contacts with colleagues, clients, or patients [2,3]. Reuter et al. [4] reported data from over 100,000 workers and demonstrated an incidence rate of 3.7 infections per 1000 workers. Alshamrani et al. [5] revealed that the risk of an infection per 100 workers (percentage) was almost ten-fold higher for healthcare workers compared with non-healthcare workers (9.78% versus 1.01%, *p* < 0.001). Mutambudzi et al. [6] concluded that both health and safety precautions, as well as the provision of personal protective equipment, are needed, particularly in the healthcare sector. As healthcare and social professionals showed a higher prevalence of SARS-CoV-2 compared to the general population [7], it is necessary to highlight this vulnerable group in post-COVID research. In Germany, 275,852 cases of COVID-19 have been recognized as occupational diseases with BK-No. 3101. Furthermore, 23,705 recognized cases of COVID-19 have been recorded as work-related accidents (according to the German Social Accident Insurance) [8]. Referring to the mentioned results, Gualano et al. [9] concluded that long- and post-COVID are rising problems in occupational medicine because they influence the return to work and quality of life in workers previously hospitalized with SARS-CoV-2. A systematic review of 13 studies confirmed that post-COVID symptoms were indicated in 16% to 87% of patients of working-age [10]. Since healthcare workers have a higher risk of infection with SARS-CoV-2 [5,11,12], the prevalence of being affected by several post-COVID symptoms may also be higher. Findings by Tabacof et al. [13] indicate a reduced full-time work ability by more than 30% after COVID-19 (median of 351 days after infection). Only half of the patients observed by Delgado-Alonso et al. [14] returned to work after a mean of 20.71 (±6.50 months) months after the infection. Of those, 32% reduced their working hours, and 23% needed a job adaptation, such as more breaks, telework, or a position change. Other studies revealed impairments in the work capacity of infected healthcare workers for nine months or longer [15,16].

Acute infection with SARS-CoV-2 can lead to several persistent complaints [17,18,19]. According to the NICE (National Institute for Health and Care Excellence) guideline, persistent signs and symptoms after an acute SARS-CoV-2 infection that cannot be justified by an alternative diagnosis are classified into either long-COVID (four weeks or longer) or post-COVID (more than 12 weeks) [20]. In the meta-analysis by Chen and colleagues [21], the prevalence of ongoing post-COVID symptoms ranged from 9 to 81% depending on, for example, virus variants, study design, study population, and measures used. The following sections demonstrate a brief overview of current research on the impacts of COVID-19 on physical and neuropsychological health under consideration of work-related SARS-CoV-2 infection. Furthermore, each section considers the importance of rehabilitation as a treatment of post-COVID symptoms. 

### 1.1. Physical Health in Long-/Post-COVID

Many COVID-19 patients have impaired physical function and decreased physical activity in the post-acute state [22,23]. Sanchez-Ramirez et al. [24] indicated that 36% of patients perceived a decrease in their functional capacity around four months after symptom onset. Results by Baricich et al. [25] demonstrate impaired physical function measured by the One-Minute-Sit-to-Stand-Test (1MSTST) in patients with COVID-19 three to six months after hospital discharge. Around 50% of patients reported functional limitations in everyday life six months after hospital discharge [26]. Age and length of hospitalization during acute infection are named as risk factors. Muscle weakness is one of the most frequent symptoms reported by hospitalized COVID-19 patients after discharge [23,27]. Paneroni et al. [28] showed that hospitalized COVID-19 patients without previous locomotor disabilities also suffered from impairment in muscle strength. These results were confirmed by Kedor et al. [29], who proved a reduction in handgrip strength in post-COVID patients. Other studies showed that COVID-19 patients suffered from impairments of their physical capacity, probably caused by muscle deconditioning [30,31]. Liao et al. [32] investigated the long-term effects of COVID-19 on healthcare workers one year after hospital discharge and showed that the physical functions and six-minute walking distance (6MWD) of healthcare workers were significantly worse compared to a healthy population. Hasenoehrl et al. [33] demonstrated significant improvements in physical performance (6MWD, 30 sec STST) in healthcare workers with post-COVID after an eight-week exercise intervention program. Increases in 30 s STST were directly associated with improvements in work ability. Initial research results indicated that rehabilitation could improve the exercise capacity of patients with acute COVID-19 and long-/post-COVID [34,35,36,37,38]. The exercise capacity of post-COVID patients, measured by 6MWD, increased after a six-week pulmonary rehabilitation program [37]. Findings by Ahmed et al. [34] supported these results in acute- and post-COVID patients. A meta-analysis also showed significant improvements in physical function (6MWD), physical activity intensity levels (Perceived Exertion Scale), and peripheral muscle performance of lower limbs (30 s STST) after pulmonary rehabilitation [39]. No study examined the effects of rehabilitation on physical performance in healthcare workers. 

### 1.2. Neuropsychological Health in Long-/Post-COVID

Cognitive impairment is another consequence of an infection with SARS-CoV-2 [40,41,42,43]. According to a systematic review, the prevalence of cognitive deficits among long-/post-COVID patients is more than 50% up to four months after acute infection. In particular, executive functions, memory, and attention differ between post-COVID patients and healthy subjects [41]. Seven to nine months after acute infection, 24–29% of patients are still affected by cognitive deficits (e.g., memory encoding/memory recall, processing speed) [44,45,46]. Omar et al. [43] showed that one to three months after infection, healthcare workers had significantly lower attention and memory scores than a healthy control group. Carazo et al. [47] confirmed these results. Crivelli et al. [41] recommended screening cognitive impairments using the Montreal Cognitive Assessment (MoCA), as this is the most commonly applied tool to detect cognitive decline. Pulmonary and other rehabilitation programs that include aerobic exercises, strength training, and educational sessions could significantly improve cognitive impairment measured using MoCA by its minimal important difference of two points [36,48].

Further, mental health impairments are frequently reported in post-COVID patients. The most common psychological symptoms one month after acute COVID-19 are anxiety and depression with a prevalence of 6–63% and 4–31%, respectively [49]. Comparable prevalence rates were found up to six months after acute infection [17,50,51]. Among healthcare workers, depression and anxiety have a prevalence of 8–44% three to ten months after acute COVID-19 [15,52,53]. Results regarding the risk factors for higher rates of anxiety and depression are inconsistent. However, some studies identified female sex or treatment in the intensive care unit (ICU) [49,50] as risk factors for anxiety and depression. Another psychological symptom associated with post-COVID is post-traumatic stress disorder (PTSD) reported by an observational study [54]. Three months after acute infection, 37% of patients had symptoms of PTSD. Despite decreasing prevalence at six months follow-up, 27% of patients still had high PTSD scores [54]. Rehabilitation programs may have a positive impact on mental health outcomes. During a three-to-six-week pulmonary rehabilitation program, the corresponding values for anxiety and depression decreased significantly [39,55,56].

Fatigue is one of the most reported post-COVID symptoms [38]. About 14 to 150 days after acute infection, the prevalence of fatigue among long-/post-COVID patients is between 33 and 58% [17,18,57,58]. According to Ceban et al. [42], fatigue symptoms decreased after 12 months but were still high with a prevalence of 32%. Haller and colleagues [59] confirmed this tendency in a population of healthcare workers suffering from post-COVID, with a prevalence of 10% for severe and 77% for mild symptoms of fatigue. Furthermore, subjective health status, hospitalization, inability to work, the severity of acute COVID, and participation in rehabilitation programs were identified as influencing factors for fatigue severity, and 40–73% of patients reported needing an appropriate rehabilitation program [59]. Results by Peters et al. [16] showed that more than 70% of healthcare workers experienced persistent symptoms, including fatigue/exhaustion, concentration/memory problems, shortness of breath, headache, and loss of sense of taste/smell for more than three months after acute COVID-19, leading to impaired work capacity for nine months or longer. The following risk factors for persistent symptoms were identified: older age, female sex, previous illness, many and severe symptoms during acute COVID-19, and outpatient medical care [16]. Delgado-Alonso et al. [14] reported a relationship between post-COVID patients who were not able to return to work and high levels of fatigue. The first results verified the positive effects of rehabilitation on fatigue [37,55,56,60]. Fatigue symptoms decreased significantly after six weeks of an outpatient pulmonary rehabilitation program [37].

In summary, the onset of several long-lasting post-COVID symptoms was confirmed by the current findings. Less is known about the impact of rehabilitation on post-COVID, especially in consideration of work-related infections, for example, of healthcare workers. Thus, the current study investigates the physical and neuropsychological health and work ability of patients who acquired COVID-19 in the workplace at the beginning and the end of inpatient post-COVID rehabilitation. Furthermore, the study aimed to detect group differences depending on sex, profession, and acute COVID status.

## 2. Materials and Methods

This study was conducted at the Chemnitz University of Technology, Germany, in cooperation with the BG Hospital Bad Reichenhall, and was registered in the German Clinical Trials Register under DRKS00022928. The study was approved by the Ethics Committee of the Bavarian State Medical Association (number 21092) and the Ethics Committee of the Chemnitz University of Technology (TU Chemnitz, Chemnitz, Germany), Faculty of Behavioral and Social Sciences (number V-427-17-KM-COVID-19-18022021). The detailed study protocol was published earlier [61]. Only information relevant to the current research question is presented here.

### 2.1. Study Design and Participants

In this longitudinal observational study with four different measurement time points, patients were recruited by a study nurse after their respective accident insurance providers registered them for rehabilitation at the German BG Hospital Bad Reichenhall. After screening for study eligibility, patients in the post-acute phase of COVID-19 as a recognized occupational disease or work-related accident signed a written informed consent form. The current study presents the results of the measurements at the beginning (T1) and end (T2) of the inpatient rehabilitation period [61]. 

All participants went through an inpatient multidisciplinary post-COVID rehabilitation program at the BG Hospital with a mean duration of 28.77 (9–42) days. In addition to medical treatment and care, patients participated in comprehensive physical and psychological treatments by specialists. For detailed information on the inpatient rehabilitation, see Müller et al. [61].

The study sample includes 127 patients (97 females/30 males, aged 21–69 years, M = 50.62, SD = 10.74) at T1. In total, 89 patients work in the healthcare sector (72 females) (Table 1), and 38 patients are classified as non-healthcare workers (e.g., administrative staff, industrial-/building technicians, social education staff, and teachers). The socioeconomic status (SES) ranged from 3.0 to 21.0 points and is divided into 3 categories: low (3.0–7.7), medium (7.8–14.1), and high (14.2–21.0) [62]. In this study, 58.5% of the patients had a medium SES and 41.5% had a high SES, while no patients were in the lowest SES category. Due to the drop-out of 3 participants at the end of rehabilitation, 124 paired samples were analyzed. One participant was confirmed to be re-infected with SARS-CoV-2 during rehabilitation and two participants were no longer interested in study participation at T2. Considering the missing values, cases for each variable vary between 112 and 124. 

### 2.2. Measurements of Sociodemographic Variables, Anamnesis, and Post-COVID Symptoms

Several sociodemographic variables (e.g., age, sex, socio-economic status, education) were obtained via questionnaire following the Questionnaires of the German Health Interview and Examination Survey for Adults (DEGS) [62,63].

The presence of post-COVID symptoms was recorded through a self-generated questionnaire (following Koczulla et al. [64]) complemented by a semi-standardized interview by a physician during medical anamnesis. Additionally, pre-existing diseases were assessed using the subscale of the work ability index (WAI) [65]. The Post-COVID-19 Functional Status (PCFS) scale was also used [66]. The PCFS scale grade ranges from 0 (no limitations/symptoms in everyday life) to 4 (severe limitations/symptoms in everyday life).

### 2.3. Physical Performance Measurements

The 6MWD was used to examine the functional exercise capacity. The minimal clinically important difference (MCID) of 30 m is recommended to determine a relevant enhancement after rehabilitation [67]. In addition to this, we recorded physical performance using the 1MSTST (MCID: ≥2) [68,69]. Quadriceps strength was measured by the isometric maximal strength test (LST) of quadriceps muscles using a functional press (Beinstemme v2, Schnell Trainingsgeräte GmbH, Peutenhausen, Germany) and the software aktivSYSTEM (aktivKONZEPTE AG, St. Ingbert, Germany). Additionally, the handgrip strength (HST) of the dominant hand was assessed with a digital grip dynamometer (JAMAR^®^ Smart Hand Dynamometer, Performance Health Supply Inc., Cedarburg, WI, USA). The subjectively perceived status of physical performance was reported by participants on a self-generated questionnaire with 10 items on a scale of 0–10 (0 = very bad, 10 = very well).

### 2.4. Neuropsychological Health Measurements

We used two German versions of the MoCA test (T1: Version 8.1, T2: version 8.2) to assess global cognitive functioning, e.g., short-term memory, visuospatial abilities, attention, concentration, working memories, language, orientation in space and time, and executive functions. Besides the calculation of the total sum score (range: 0–30 points), participants were classified into one of two groups. Subjects with a score of 25 or lower are considered to be mildly cognitively impaired (MCI). A MoCA score between 26 and 30 is classified as cognitively healthy (CHI) [70]. The MCID for the MoCA is a change of two points in the total sum score [71]. Furthermore, the Digital Simple Substation Test (DSST) was used to examine various cognitive functions, including sustained attention, visual–spatial skills, response speed, and set-shifting. Participants were given a sheet of paper with rows of symbols and the task of matching each symbol to a number. A legend at the top of the page indicated which symbol matches which digit (1–9). Afterward, the number of correct digit-symbol matches executed in 90 s was recorded [72].

The German version of the Hospital Anxiety and Depression Scale (HADS-D) was used to assess the presence of psychological symptoms. The subscales for depression (HADSdepression) and anxiety (HADSanxiety) consist of seven items, respectively. A sum score for each scale was generated (range 0–21) based on a Likert scale from 0 (‘no symptoms’) to 3 (‘severe symptoms’). Scores between 8 and 10 indicated mild symptoms, while scores above 14 indicated moderate to severe symptoms [73,74]. Based on analyses with cardiac patients after rehabilitation, an MCID of >1.7 points was reported in each subscale [75].

Two different self-administered fatigue questionnaires were administered. The Brief Fatigue Inventory (BFI) consists of ten questions concerning fatigue and fatigue-related symptoms to detect fatigue severity. The mean score of all items ranges from 0–10, a higher score indicates a more severe level of fatigue [76,77]. The Fatigue Impact Scale (FIS) includes 40 items assigned to the three subscales (cognitive, physical, and psychosocial functioning). A sum score is calculated (range: 0–160) after answering all questions on a five-point Likert scale from 0 (‘no problem’) to 4 (‘extreme problem’). Higher scores indicate more severe functional impairments due to fatigue [78].

The subjectively perceived status of neuropsychological health was reported by participants on a self-generated questionnaire with 10 items on a scale of 0–10 (0 = very bad, 10 = very well).

### 2.5. Employment and Work Ability Measurements

The ability to work was assessed using the Work Ability Index (WAI) questionnaire [65]. The WAI is a widely used questionnaire that considers the worker’s health status, demands of work, and resources. It is composed of seven subscales that correspond to one or more questions: (1) current work ability compared to the best possible workability over the lifetime, (2) ability to work in relation to job requirements, (3) number of diseases diagnosed by a physician, (4) estimated loss of work ability because of illness, (5) absence from work in the previous year, (6) subject’s own prognosis of work ability, and (7) mental resources. The index is calculated by the sum of the points on each item. Scores between 7 and 27 points indicate poor, 28–36 moderate, 37–43 good, and 44–49 excellent work ability [79]. Patients also reported occupational inability caused by COVID-19 and post-COVID-19 prior to rehabilitation.

### 2.6. Statistical Analyses

Data were analyzed using SPSS software (version 29, SPSS Inc., Armonk, NY, USA). As most of the parameters were not normally distributed, the Wilcoxon signed-rank test was used to compare variables at T1 and T2. Group differences relating to sex (male vs. female), profession (profession in healthcare services vs. other), and acute COVID-status (mild/moderate COVID-19 vs. severe/critical COVID-19) at T1 and T2 (Δ = MT2 − MT1) were analyzed using the Mann–Whitney U test. Missing data were noted and are presented clearly in the Tables, and *p*-values of < 0.05 were considered statistically significant. Effects sizes were reported as r. According to Fritz, Morris, und Richler [80], an effect size r of 0.1 represents a ‘small’ effect size, 0.3 a ‘medium’ effect size, and 0.5 a ‘large’ effect size.

## 3. Results

The current study investigated changes in physical and neuropsychological health and work ability at the end of inpatient rehabilitation.

### 3.1. COVID-19 Infection, Risk Factors, and Post-COVID Symptoms

At the beginning of rehabilitation, participants had been infected with SARS-COV-2 a mean of 408.81 (range: 124–813) days prior (Table 2). Thus, all patients were infected with virus variants alpha or delta [81]. Depending on the WHO classification for the severity of COVID-19, 91 patients showed a mild or moderate course of acute COVID-19, while 36 patients were classified as severe or critical, and 33 were hospitalized for a mean duration of 14.10 (range: 1–100) days (Table 2). In total, 10 patients were treated in an ICU for a mean duration of 10 (5–21) days. During hospitalization, 27 patients needed oxygen therapy and 6 patients needed mechanical ventilation. 

Participants showed a mean body mass index of 31.47, and 85.2% were classified as overweight or obese according to the WHO. In total, 8.7% of participants were current smokers, and 38.6% were former smokers (Table 1), and 95.3% of the patients had comorbidities (e.g., hypertension, coronary heart disease, chronic bronchitis, asthma, diabetes mellitus, neurological diseases, oncological diseases) prior to contracting COVID-19 (Table 2).

All participants showed ongoing symptoms related to post-COVID at T1. Figure 1 illustrates 13 summarized symptom clusters, while a detailed recording of symptoms is presented in Appendix A in the Appendix A. All patients indicated having symptoms of exercise intolerance, 98% suffered from neurological ailments, 91% from fatigue, and 91% from chest pain. As seen in Figure 1, there was a decrease in the prevalence of all symptoms at T2 except for fatigue.

The PCFS score indicated that 9.6% of the patients had no functional limitations (grade 0) and 2.4% had negligible functional limitations (grade 1) at the beginning of rehabilitation. In total, 34.4% of the patients reported slight functional limitations (grade 2), 52.0% moderate functional limitations (grade 3), and 1.6% severe functional limitations (grade 4). The PCFS scores had decreased significantly by the time patients were discharged from rehabilitation (*p* = 0.017). A total of 17.8% of the patients demonstrated grade 0, and 3.4% demonstrated grade 1. Most of the patients had slight (26.3%) or moderate functional limitations (51.7%) at T2. Only 0.8% of the patients had severe functional limitations at the end of rehabilitation (grade 4).

### 3.2. Group Differences at the Beginning of Rehabilitation in Comparison to Sex, COVID-Status, and Employment

#### 3.2.1. Physical Health

For physical performance measurements, no significant differences were observed between the mild/moderate COVID-19 group and the severe/critical COVID-19 group. The median of the handgrip strength was significantly different between males (38.35, interquartile range (IQR): 32.38–49.12) and females (25.43, IQR: 20.08–31.32, *p* < 0.001). Quadriceps strength differed significantly at the beginning of rehabilitation, with a median of 134.99 (IQR: 115.54–145.81) for male patients and 85.19 (IQR: 65.23–113.05, *p* < 0.001) for female patients. In addition, the median quadriceps strength for healthcare workers (88.82, IQR: 67.10–120.97) differed significantly from non-healthcare workers (124.16, IQR: 83.62–141.38, *p* = 0.005). No significant differences in handgrip strength were found for the respective professions. Detailed results are presented in Appendix A.

#### 3.2.2. Neuropsychological Health

The MoCA score (27, IQR: 26–29) for the mild/moderate COVID-19 group was significantly higher compared to patients with a severe/critical course of disease (26.5, IQR: 25–28, *p* = 0.009). The DSST showed similar results, with a median of correct number-symbol matches of 47 (IQR: 39–54) for the mild/moderate COVID-19 group and 41.5 (IQR: 34–51.5, *p* = 0.022) for the severe/critical COVID-19 group. The remaining neuropsychological health outcomes did not show any significant difference at T1 between these two groups (Appendix A). The DSST showed that cognitive function at baseline differed significantly between males and females, with a median of 41 (IQR: 34.5–47.5) for male patients and 47 (IQR: 38.5–54, *p* = 0.009) for female patients (Appendix A). The median FIS score for male patients was significantly lower, with a median of 83.5 (IQR: 60.3–110.5) than for females, with a median of 99 (IQR: 77.5–115, *p* = 0.029). Finally, no significant difference between professions was observed for any of the neuropsychological parameters at T1 (Appendix A). 

#### 3.2.3. Work Ability

Due to COVID-19, 86 participants were not able to work at the beginning of rehabilitation (Table 1). No significant difference was found between the severity of acute COVID-19, sex, and professions at T1 for either the subscores or the total WAI score. Detailed results are presented in Appendix A.

### 3.3. Rehabilitation Outcomes

#### 3.3.1. Physical Health

Nearly all physical performance measurements improved significantly at the end of rehabilitation (Table 3). Between T1 and T2, the median of the 6MWD increased by 64 m (IQR: 32.00–112.00, *p* < 0.001). Overall, 90 patients showed improved performance during 6MWD of 30 m or more, achieving MCID (Holland et al. 2014), while 19 patients improved below the threshold of 30 m. Ten patients did not improve 6MWD. Physical performance, measured using 1MSTST, increased from a median of 20 (IQR: 16–24) to 22 (IQR: 17–26, *p* = 0.001). Seventy patients achieved the MCID of at least two more repetitions [68]. Furthermore, patients improved their quadriceps strength by 12.80 kg (IQR: 0.51–29.19, *p* < 0.001). Handgrip strength increased from a median of 27.63 (IQR: 20.63–35.07) to 29.47 (IQR: 21.02–36.17), but the change was not significant. After rehabilitation, the subjectively perceived status of physical performance increased from a median of 4.67 (IQR: 3.44–6.11) to 5.78 (IQR: 4.56–7.22, *p* < 0.001).

Furthermore, no significant difference over time was observed for the group-wise comparison between males and females, healthcare workers and non-healthcare workers, or mild/moderate COVID-19 patients and severe/critical COVID-19 patients in terms of physical performance (see Appendix A).

#### 3.3.2. Neuropsychological Health

Neuropsychological health outcomes improved significantly after post-COVID rehabilitation (Table 3). At T1, the median of the MoCA test was 27 (IQR: 25–28). Mild cognitive impairment was found in 24.8% of patients. A significant improvement in cognitive function was found at the end of rehabilitation, with a median MoCA score of 27 (IQR: 26–29, *p* = 0.015). The prevalence of patients with mild cognitive impairment decreased by ~4% to 21.3% at T2. The DSST is consistent with this result, as the improvement was statistically significant (*p* < 0.001). At T1, 22% of patients had mild symptoms of depression and 28% showed symptoms of severe depression. The score of HADSdepression decreased significantly from a median of 7 (IQR: 4–11) to 6 (IQR: 3–10, *p* < 0.001). For anxiety, 21% had mild symptoms and 25% had severe symptoms at T1 and the median score for HADSanxiety decreased from 7 (IQR: 4–11) to 5 (IQR: 3–10, *p* < 0.001). At baseline, 60.6% of patients showed moderate and 24.4% severe symptoms of fatigue according to the BFI score (T1: 5.6, IQR: 4.6–6.7). Similar to depression and anxiety, the scores for fatigue declined significantly in both measurement tools. The FIS score decreased significantly from a median of 97 (IQR: 73–113) to 85.5 (IQR: 65–111.75, *p* = 0.001), and the BFI decreased from 5.6 (IQR: 4.6–6.7) to 5.3 (IQR: 3.9–6.6, *p* = 0.004). At rehabilitation discharge, the prevalence of fatigue (BFI score) was 53.0% for moderate and 23.1% for severe symptoms. Furthermore, the subjectively perceived status of neuropsychological health also improved significantly (T1: 5.3, IQR: 4.5–6.6; T2: 5.8, IQR: 4.5–7.3, *p* < 0.001). Regarding the group-wise comparison of the change in neuropsychological parameters over time, only the HADSdepression score reached a significant result between the two professional groups. The score for the group of healthcare workers decreased by a median of –1 (IQR: −3–0), whereas the median decrease of the non-healthcare workers was 0 (IQR: −1–1, *p* = 0.027). For a detailed overview of the group-wise comparison see Appendix A.

#### 3.3.3. Work Ability

At baseline, 86 patients (healthcare workers: 69.8%) were unable to work due to COVID-19. After rehabilitation discharge, 90 out of 124 patients were still unable to work, 73.3% of whom were healthcare workers (Table 1). No significant difference was found between T1 and T2 in any subscale of the work ability index. In addition, the total score of the WAI between T1 (24.75, IQR: 21–28) and T2 (24.75, IQR: 21–28) did not increase significantly (*p* = 0.408). At baseline, 74.0% of the patients reported a poor and 26.0% a moderate work ability. At T2, 69.8% of the patients reported poor, 29.3% moderate, and 0.9% good work ability. Further, no significant difference over time was observed for the group-wise comparison between sex, professions, and severity of acute COVID-19 according to the WAI. All results and corresponding effect sizes are given in Appendix A.

## 4. Discussion

To date, this study is one of the first in Germany to examine work ability next to physical performance and neuropsychological health outcomes following inpatient post-COVID rehabilitation of patients who acquired COVID-19 in the workplace. In summary, physical and neuropsychological health were improved after rehabilitation, but no change in work ability could be confirmed. The role of sex, the severity of acute COVID-19, and employment were also considered in all analyses.

### 4.1. Post-COVID Symptoms

Referring to long-lasting symptoms after acute COVID-19, our results at the beginning of post-COVID rehabilitation were similar to those of other current studies [16,38]. As in the current study, exercise intolerance, neurological ailments, fatigue, pain, and psychological symptoms are the most commonly reported long-term symptoms of post-COVID in previous studies [82,83]. Han et al. [84] and Mazza et al. [85] confirm the current finding that COVID-19 survivors continue to experience post-COVID symptoms, such as fatigue, depression, and anxiety, even after one year. D’Ettorre et al. [10] showed an incidence between 16 and 87% for post-COVID in working-age patients, and up to 70% for healthcare workers [16,53,86]. The systematic review by Soril and colleagues [87] confirmed that psychological post-COVID symptoms (dyspnea, fatigue, anxiety, and depression) improved after several pulmonary rehabilitation programs. In our study, the prevalence of most symptoms decreased after rehabilitation, whereas there were persistent physical, psychological, and cognitive dysfunctions. This is in line with the results from Brehon et al. [88], who showed that neuropsychological symptoms improved in healthcare workers after a rehabilitation program. As fatigue is one of the most commonly reported post-COVID symptoms [38] and the reported prevalence of fatigue at T2 remains high, sustainable treatments and therapies addressing fatigue must be developed and implemented.

### 4.2. Physical Health

At the beginning of rehabilitation, patients showed no significant differences in 6MWD and 1MSTST dependent on sex, the severity of acute COVID-19, or employment. In contrast, Gloeckl et al. [36] reported a lower 6MWD in patients with severe/critical COVID-19. This could be explained by the lower time interval for hospital discharge and admission to rehabilitation (5–40 days) compared to our study (124–813 days). As expected, the sex-based differences in hand and quadriceps strength were based on the different constitutional conditions of men and women [89]. After rehabilitation, physical performance increased with small to large effect sizes, except for handgrip strength. Performance at 6MWD improved by an average of more than twice the MCID. Our result is comparable to the mean difference of 65.85 m reported in the meta-analysis with randomized controlled studies by Ahmed and colleagues [34]. Moreover, the MCID of 1MSTST was achieved at T2. This is in line with results from other current studies [34,36,37,56,90]. While the muscle strength of the legs increased significantly with a large effect size, the improvement of handgrip strength was not significant over time in our study population. While current studies confirm the improvement of quadriceps strength after rehabilitation [36,91], the results of the change of hand force after rehabilitation are inconsistent. Significant improvements were observed in some studies [34,36], whereas other studies [91,92] demonstrated only trends comparable with our results. Muscle fatigue of handgrip is part of fatigability. As 70% of the patients were categorized as moderately to severely fatigued at T2, this could be one reason for the lack of improvement in handgrip strength. Both mild/moderate and severe/critical COVID-19 patients benefitted from post-COVID rehabilitation regarding physical performance, as also reported by Ahmed et al. [34]. Following Rodriguez-Blanco et al. [93], our study found no differences between the sexes related to the improvement in physical capacity. Liao et al. [32] showed that the physical functions and 6MWD of healthcare workers were significantly worse compared to a healthy population. Hasenoehrl et al. [33] demonstrated significant improvements in the 6MWD of healthcare workers with mild to severe long-COVID (57.6–68.9 m) after an eight-week exercise intervention program. Our result is comparable with the reported improvement of 77.50 m among healthcare workers after rehabilitation.

Next to the previously mentioned outcomes, the subjective perception of individual physical performance also increased after rehabilitation. Improved body awareness is required to recognize individual physical performance limits, especially in the simultaneous occurrence of fatigue and post-exertional malaise. Consequently, improved body awareness is important for aftercare and long-term maintenance of physical performance.

When discussing our results against the current literature, the natural recovery process of the disease and the different study designs used (sample size, date of rehabilitation after acute COVID-19, length of rehabilitation, contents of inpatient/outpatient rehabilitation) must be considered [35,39]. Nevertheless, systematic reviews and meta-analyses investigating COVID-19-related physical activity-based rehabilitations confirmed significant increases in functional capacity, quality of life, as well as mental health [34,35]. Due to the known potential positive effects of physical activity on physical and psychological health in the general population or in patients with non-communicable diseases [94], regular physical activity should be implemented in daily life to maintain the benefits of rehabilitation in the long term. The individual level of physical performance should be considered. In addition, long-/post-COVID self-help support groups can have an important influence on maintaining rehabilitation effects [64].

### 4.3. Neuropsychological Health

The highly significant improvement in subjectively perceived mental health after rehabilitation discharge is a first indication of the successful implementation of the inpatient rehabilitation program with regard to neuropsychological parameters. At rehabilitation admission, almost 25% of patients were classified as mildly cognitively impaired. The observed prevalence agrees quite well with the prevalence after six months of infection in the study by Hartung et al. [45]. The group-wise comparisons at T1 resulted in significantly better scores for the MoCA test and DSST in patients with mild to moderately acute COVID-19. Despite this difference, both groups benefitted from the rehabilitation program to the same extent. The significant improvement in the cognitive parameters with small to moderate effects indicates that the inpatient rehabilitation program can reduce the cognitive deficits of post-COVID patients. The current study confirms the outcome of Daynes et al. [48]; although, their demonstrated improvement of two points in the MoCA score was higher than in our study. We presume that the better baseline scores of the current study population made it difficult to reach greater improvements.

At the baseline measurement, the total prevalence of HADSdepression and HADSanxiety was 50% and 46%, respectively. These results are within the same range as other long-term studies [15,52], indicating a prevalence of depression and anxiety among healthcare workers of up to 44%. In the current study, the included participants were also mostly employed in the healthcare sector. At rehabilitation discharge, HADSdepression and HADSanxiety scores were reduced significantly, with a medium effect size. This result confirms the outcomes of previous studies [55,56]. The remaining prevalence after rehabilitation discharge for mild to severe symptoms of depression and anxiety was 37% and 36%, respectively. Interestingly, the longitudinal comparison of healthcare and non-healthcare workers revealed a significantly stronger reduction of symptoms of depression in patients working within the healthcare sector. This highlights how responsive this vulnerable occupational group was to the inpatient rehabilitation program. A 2022 report by one of the main health insurance companies of Germany (‘DAK-Gesundheit’) shows that healthcare workers have the highest duration of incapacity to work due to psychological diseases of all occupational groups [95]. This may be due to the working conditions, environment, and lack of adequate supervision. These circumstances should be taken into consideration when assessing the efficacy of the rehabilitation program.

The prevalence of moderate to severe symptoms of fatigue at T1 was 85%, according to the assessed BFI score. The observed prevalence is more than twice as high as the result of a previous study [42]. One explanation for this result could be the characteristic of the current study sample. All participants were diagnosed with COVID-19 recognized as an occupational disease or work-related accident. Furthermore, the course of the disease was quite long, at ~409 days. Similar to Ceban and colleagues [42] and Haller and colleagues [59], female patients showed significantly higher FIS scores at timepoint T1 than male patients, with the same tendency for the BFI score. After rehabilitation discharge, the FIS and BFI scores improved significantly, and the prevalence of fatigue was reduced by 8%. The study of Hayden et al. [56] also revealed a decline in fatigue after three weeks of pulmonary rehabilitation, although to a greater extent. It is important to note that the period between infection and rehabilitation was around six times shorter than in the current study. Although Hayden et al. [56] showed a higher efficacy for patients with an earlier start of rehabilitation, the current data suggest that patients with a longer latency between the disease and rehabilitation can also benefit from the interventions.

Evidence suggests that fatigue, cognitive symptoms, and depressive symptoms interact with each other. A study with a similar fatigue prevalence eight months after acute COVID-19 demonstrated that depressive symptoms and cognitive deficits can predict overall fatigue [96]. Further, at a nine-month follow-up, Mirfazeli et al. [97] showed an association between neuropsychiatric symptoms (e.g., depression), lower MoCA scores, and fatigue. The interaction between these three symptoms in post-COVID patients, which is still not fully understood, emphasizes that the diversity of symptoms does not occur independently, but that there may be a common mechanism behind these sequelae. Peripheral inflammation processes can influence the performance of cognitive tasks [98] and lead to higher rates of fatigue [45]. Further, observed brain damage due to the SARS-CoV-2 infection is another explanation for the neuropsychological impairments [45,99]. Lastly, depressive symptoms could be induced by cytokine storms and increased activation of microglia [100,101,102].

In summary, the results of the neuropsychological parameters encourage the implementation of post-COVID rehabilitation programs for post-COVID patients not only to improve physical health outcomes but also to address cognitive and mental health impairments. According to the persistently high rates of depression and anxiety after rehabilitation, post-COVID patients should be encouraged to seek ongoing outpatient psychological treatment after rehabilitation discharge to secure ongoing treatment of the reported mental impairments.

### 4.4. Work Ability

In our study, 86 out of 127 patients were unable to work at the beginning of the rehabilitation program, 69.8% of whom were healthcare workers. This is in line with other studies, demonstrating that even a mild COVID-19 infection may lead to a substantial reduction in work ability [103], especially in healthcare workers [104]. After rehabilitation discharge, 90 out of 124 patients were still unable to work. Brehon et al. [88] confirm this result and indicate that only 53% of workers with post-COVID symptoms returned to work after a rehabilitation program. In our study, many healthcare workers (60 out of 89) were incapacitated for work at baseline. This result could be due to the characteristics of the working conditions (e.g., psychological stress in the healthcare sector during the COVID-19 pandemic, workload, and work schedule) and the persistence of COVID-19 symptoms, especially neuropsychological symptoms. As 70% of the patients were categorized as moderately to severely fatigued at rehabilitation discharge, this could be a reason for limited work capacity. Results by Delgado-Alonso et al. [14] support this and found an association between incapacity to work, high levels of fatigue, and low cognitive performance. Furthermore, the high prevalence of physical symptoms (100% exercise intolerance; 81% joint or muscle pain) reported by patients at T1 and the physical demands of working in the healthcare sector could result in a lack of work capacity. Findings by Pauwels et al. [105] indicate that return to work varies individually with persistent COVID-19 symptoms.

To the best of our knowledge, this is the first study in Germany to assess work ability, as measured by the WAI, before and after rehabilitation in patients with post-COVID-19. After rehabilitation discharge, 69.8% of the patients reported poor, 29.3% moderate, and only 0.9% good work ability. Andrade et al. [106] examined the work ability among Brazilian workers (N = 1211) at baseline and twelve months follow-up. The authors found that nearly 75% of the workers with COVID-19 reported a good to excellent work ability at baseline (mean WAI: 39.2) and at a 12-month follow-up (mean WAI: 39.0). No difference in WAI scores was found between infected and uninfected workers. The authors concluded that these findings could be explained by job stability and working from home as protective factors because many of the patients were public servants. Most of our sample is composed of healthcare workers with greater job demands. This assumption may explain our results, which, in contrast to Andrade et al. [106], showed a poor work ability at baseline (WAI median: 24.75, IQR: 21–28) and after rehabilitation discharge (WAI median 24.75, IQR: 21–28).

At the beginning of rehabilitation, there were no significant differences in the total WAI scores between the mild/moderate COVID-19 group and the severe/critical COVID-19 group, between males and females, or between healthcare workers and non-healthcare workers. In contrast, other studies found that the work ability and return-to-work differed for sex and the severity of COVID-19 infection [9,58,107]. Our study only included patients registered for rehabilitation by their respective accident insurance companies, which is possibly why many patients were restricted in their ability to work and no differences between the subgroups were observed. The high rate of inability to work and the high amount of post-COVID symptoms support Böckermann et al. [108], demonstrating that poor health status is linked with a higher rate of unemployment. To increase return-to-work among post-COVID patients, it is necessary to adapt the working conditions (adjustments in workload, working hours, and tasks) and develop a return-to-work plan [88,109]. Strategies supporting the return to work after COVID-19 could be similar to programs developed for chronic diseases [110,111,112].

The number of workers unable to return to work due to COVID-19 will have a huge impact on the labor force around the world. This applies, in particular, to occupational fields in the health sector, which are of great importance in times of a pandemic. Outcomes related to work capacity should be recorded during clinical trials and studies evaluating interventions for patients with long or post-COVID [113]. Furthermore, rehabilitation programs designed to improve physical and neuropsychological health may indirectly help to improve the ability to work.

### 4.5. Limitations

The results should be interpreted carefully under consideration of an observational study design without a control group. Thus, the impact of natural recovery should also be addressed. We agree with Gloeckl and colleagues [36], that the improvement of physical performance and neuropsychological health is more likely associated with rehabilitation than natural recovery. To date, there is only a limited number of randomized controlled studies analyzing rehabilitation outcomes in patients with post-COVID [35,38,114]. Therefore, future randomized controlled studies are essential to confirm the evidence of the benefits and effects of post-COVID rehabilitation.

It must be mentioned that the sample sizes of the three different subgroups relating to sex, profession, and acute COVID status were not well-balanced. For example, we included more women in our study. This circumstance may be due to more women working in healthcare services than men [115]. Further studies with similar sex ratios are needed to detect sex-based differences in the effects of rehabilitation.

Furthermore, we investigated a selective study population with work-related SARS-CoV-2 infection, e.g., healthcare workers with special professional circumstances compared to the participants of other studies in the context of rehabilitation [36,56]. Healthcare workers reported higher suicidal ideation, increased stress, and decreased quality of life during the COVID-19 pandemic than other professionals [116]. This must be considered in comparison with findings from other studies in the context of COVID-19 rehabilitation.

Given these circumstances, it should be noted that a convenience sample was recruited, which was supposed to comprise at least 115 cases including a drop-out rate of 25% after preliminary power calculation, as mentioned in the detailed study protocol [61].

## 5. Conclusions

Despite the high prevalence of persistent post-COVID symptoms in this study population, improvements in physical performance and neuropsychological health after inpatient multidisciplinary post-COVID rehabilitation were detected. These results confirm the strengths of this specific program. The study shows that 72.5% were still unable to work after rehabilitation discharge, and most patients demonstrated poor or moderate work ability. Thus, to further improve the ability to work further, it is necessary to focus more on individual causes of work disability and to address these during inpatient rehabilitation, but also to continually improve aftercare strategies. Our study is limited to two measurement points to analyze the immediate health outcomes before and after rehabilitation. Analyzing the aftercare process is also necessary to evaluate the benefits of rehabilitation in the long run. Therefore, further investigations at 6 and 12 months after rehabilitation could examine the further course of post-COVID regarding health and work ability status. Taken together, the described findings can confirm the feasibility and the efficacy of post-COVID rehabilitation compared to the results of the systematic review by Baily and colleagues [35].

## Figures and Tables

**Figure 1 ijerph-20-01468-f001:**
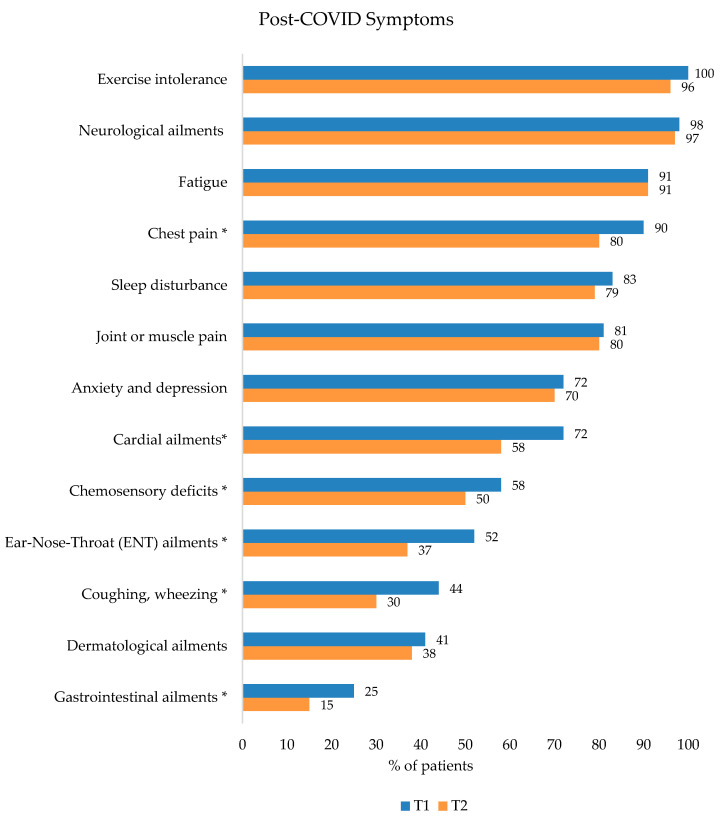
Summarized symptom clusters of post-COVID patients before (T1, blue) and after (T2, orange) rehabilitation. Significant differences are marked with * (*p* < 0.05).

**Table 1 ijerph-20-01468-t001:** Characteristics of the study population (N = 127).

	N (%)	Mean	SD	Min	Max
Sex female					
• male	30 (23.6)				
• female	97 (76.4)				
Age [years]		50.62	10.74	21	69
BMI [kg/m^2^]		31.47	6.61	19.1	54.5
• Normal	19 (14.8)				
• Overweight	41 (32.0)				
• Obesity class I	36 (28.3)				
• Obesity class II	20 (15.6)				
• Obesity class III	11 (8.6)				
Smoking status					
• Currently	11 (8.7)				
• Former	49 (38.6)				
• Never	67 (52.7)				
Healthcare workers	89 (70.0)				
• Nursing staff	71 (55.9)				
• Medical staff	7 (5.5)				
• Paramedical staff	6 (4.7)				
• Therapeutic staff	5 (3.9)				
Non-Healthcare workers	38 (30.0)				
• Administrative staff	11 (8.7)				
• Industrial-/Building technicians	9 (7.1)				
• Social educational staff and teachers	7 (5.5)				
• Cosmeticians	3 (2.4)				
• Retail salesman/-woman	3 (2.4)				
• Personal service staff	3 (2.4)				
• Other occupations	2 (1.6)				
Occupational inability					
• Occupational inability prior to rehabilitation [days]		135.27	153.85	6	544
• Pre rehabilitation o Healthcare workers (n = 89) o Non-healthcare workers (n = 38)	86 (67.7) 60 (69.8) 26 (30.2)				
Post rehabilitation (N/A = 3) o Healthcare workers (n = 87) o Non-Healthcare workers (n = 37)	90 (72.5) 66 (73.3) 24 (26.7)				

N/A = not available, no answer. SD = standard deviation.

**Table 2 ijerph-20-01468-t002:** Course of COVID-19 disease and comorbidities prior to COVID-19.

	N (%)	Mean	SD	Min	Max
Hospitalization	33 (25.9)				
• Duration of hospitalization [days]		14.10	19.01	1	100
ICU stay	10 (7.9)				
• Duration of ICU stay [days]		10.83	5.46	5	21
Disease severity according to WHO					
• Mild/moderate	91 (71.7)				
• Severe	28 (22.0)				
• Critical	8 (6.3)				
O2 therapy during hospitalization	27 (21.2)				
Mechanical ventilation	6 (4.7)				
Duration between first positive PCR test and rehabilitation admission [days]		408.81	140.89	124	813
Duration of rehabilitation [days]		28.77	5.44	9	42
Comorbidities prior to COVID-19					
• Hypertension	38 (29.9)				
• Coronary heart disease	5 (3.9)				
• Chronic bronchitis	5 (3.9)				
• Asthma	21 (16.5)				
• Diabetes mellitus	10 (7.8)				
• Neurological diseases	9 (7.1)				
• Oncological diseases	14 (11.0)				

N/A = not available, no answer. SD = standard deviation.

**Table 3 ijerph-20-01468-t003:** Outcomes of an inpatient rehabilitation program in post-COVID patients.

	N	Pre Median (IQR)	Post Median (IQR)	Δ	z	*p*	r
Physical Performance							
6MWD [m]	119	520.00 (447.00–570.00)	576.00 (522.00–636.00)	64.00 (32.00–112.00)	−9.032	<0.001	−0.828
1MSTST	118	20 (16–24)	22 (17–26)	2 (−1–5)	−4.960	0.001	−0.456
Handgrip strength [kg]	121	27.63 (20.63–35.07)	29.47 (21.02–36.17)	−0.17 (−2.95–4.02)	−0.960	0.337	−0.087
Quadriceps strength [kg]	120	95.43 (69.36–131.83)	110.13 (88.49–146.42)	12.80 (0.51–29.19)	−6.973	<0.001	−0.637
Subjective physical ability	112	4.67 (3.44–6.11)	5.78 (4.56–7.22)	1.06 (0.22–2.22)	−6.148	<0.001	−0.581
Neuropsychological Parameters							
MoCA score	122	27 (25–28)	27 (26–29)	0 (−1–2)	2.434	0.015	0.220
DSST	122	46 (37–53)	50 (40–57)	2.5 (−1–6)	4666	<0.001	0.422
HADSdepression	122	7 (4–11)	6 (3–10)	−1 (−3–0)	−4.477	<0.001	−0.405
HADSanxiety	122	7 (4–11)	5 (3–10)	−1 (−3–0)	−4.444	<0.001	−0.402
FIS	120	97 (73–113)	85.5 (65–111.75)	−5 (−16–4)	−3.262	0.001	−0.297
BFI	122	5.6 (4.6–6.7)	5.3 (3.9–6.6)	−0.21 (−0.99–0.43)	−2.848	0.004	−0.257
Subjective mental health	122	5.3 (4.5–6.6)	5.8 (4.5–7.3)	0.31 (−0.39–1.26)	3747	<0.001	0.339
Work ability							
WAI score	115	24.75 (21–28)	24.75 (21–28)	0 (−2–2)	−0.827	0.408	−0.077
• Work ability	118	3 (0–6)	3 (0.75–6)	0 (−1–2)	−1.787	0.074	−0.165
• Work requirements	117	7 (6–9)	7 (6–8)	0 (−1–1)	−2.584	0.010	−0.239

IQR = interquartile range.

## Data Availability

The data are available from the corresponding author upon request.

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
