# Peer review of "Impact of Rehabilitation on Physical and Neuropsychological Health of Patients Who Acquired COVID-19 in the Workplace"

_ijerph, 2023, doi:10.3390/ijerph20021468_

Round 1
Reviewer 1 Report
I think it is really an important topic to the post COVID period. The utility and efficacy of the rehabilitation may play a crucial role in returning to daily life. It seems that the program presented in the article has a positive influence, thus a discussion of strengths and weaknesses, based on the study results, might be helpful.
The research is well described, however a graphical abstract or another figure showing the outcome of the program can help visualize it in a better way
Author Response
Thank you for reviewing our manuscript and your helpful comments. We have critically revised the manuscript. Please find below our answers and changes made in the manuscript to each of your comments. We hope we have sufficiently incorporated your feedback into the new version of the manuscript.
To visualize the outcome of the study in a better way, we added a graphical abstract (see attachment). You are right, that a discussion of the rehabilitation program might be helpful. Therefore, we added the following sentence (page: 15, line: 691-698): “Despite the high prevalence of persistent post-COVID symptoms in this study population, improvements in physical performance and neuropsychological health after inpatient multidisciplinary post-COVID rehabilitation were detected. These results confirm the strengths of this specific program. The study shows that 72.5% were still unable to work after rehabilitation discharge, and most patients demonstrated poor or moderate work ability. Thus, to further improve the ability to work further, it is necessary to focus more on individual causes of work disability and to address these during inpatient rehabilitation, but also to continually improve aftercare strategies.”

Reviewer 2 Report
The main aim of the study is to examine physical and neuropsychical health and work ability of patiens who acquired COVID 19 in the workplace. The study also aimed to detect group differences depending on sex, profession, and acute COVID-status.
The study is significant within the field, because I aims to fill the gap in the post-COVID 19 research. The focus of the study is on the impact of rehabilitation on COVID work-related infections. There is a disctinct lack of research conducted on the effects of rehabilitation on physical performance in healthcare workers.
Thanks to its longitudinal dataset, the study is different from other research, which are usually limited to pandemic data alone. In addition, less is known about the effect of rehabilitation on infected groups. The study is designated to examine the impact of rehabilitation. It helps to improve strategies in aftercare concerning health and work ability status.
The study design and sample are well described. I have confidence in statistical analyses. The authors acknowledge certain study limitations and their explanations sound legitimate to me. The authors are fully aware that randomized controlled studies are essential to confirm the evidence of the benefits and effects of the post-COVID rehabilitation. Therefore, the results of the study should be considered carefully. There is also a question of natural recovery and its impact on patiens. The sample of the study needs to be revised in order to discuss possible sex-based differences in effects of rehabilitation. It seems that there is also a need in taking into account work-related characteristics, because of the main focus of the study on healthcare workers and their exposure to risk during the COVID 19 pandemic.
Nonetheless, the study offers an important insight into physical performance and neauropsychological health in the light of further research. The conclusion is consistent with the evidences presented in previous parts of the article. In my opinion, the authors achieved the aim of their study.
The literature review includes references to previous topic-related research from well-established academic journals. It also provides necessary background information for the study. All tables and figures are good quality and readability.
Author Response
Thank you for reviewing our manuscript and your helpful comments. We have critically revised the manuscript. Additionally, a native English speaker has proofread the new version of the manuscript. We hope we have sufficiently incorporated your feedback into the new version of the manuscript.
Reviewer 3 Report
Currently, any new data on post/long-COVID symptoms and the effects of rehabilitation are desirable. This is well designed study (despite of lack of the control group that is difficult, even impossible, to gain in this case).
Author Response

(The authors gave the same response as above.)

Reviewer 4 Report
This study focused on the impact of rehabilitation on health outcomes associated with post-COVID of people who work in healthcare field. The physical capacity; neuropsychological health work ability are index to measure the outcome in this study. This study provided a comprehensive analysis and a different angle for the post-COVID outcome in work force after rehabilitation. However, further investigations of this study population for six and 12 months after rehabilitation is ongoing.
Suggestions:
There is still some information could be analyzed based on the data. For example in 3.1. COVID-19 Infection, Risk Factors, and Long-/Post-COVID symptoms section, it would be good to see the relationship between the recovery of symptom groups that relate to the change of physical capacity; neuropsychological health and work ability. It could be provided as an indicator to detect the recovery of work force.
To make it clear, the data showed the post-COVID symptoms with significant change between T1 and T2 in Figure 1 are coughing, wheezing / ear-nose-throat (ENT) ailments / chemosensory deficits / gastrointestinal / cardinal ailments/ chest pain. Those symptoms I would say are short-term/chronic symptoms that are largely recovered during your study period. By analyzing and combining the data in Figure 1 and Table3, the recovery of those symptoms would by what degree to restore the physical capacity; neuropsychological health work ability?
Author Response
Thank you for reviewing our manuscript and your helpful comments. Please find below our answers to your comments. We have critically revised the manuscript. Additionally, a native English speaker has proofread the new version of the manuscript. We hope we have sufficiently incorporated your feedback into the new version of the manuscript.
Thank you very much for your suggestion. You are absolutely right. At the moment we are writing a further manuscript taking into account the associations of several parameters at the beginning and the end of rehabilitation as mentioned in the related study protocol. Referring to the current literature [1,2], we will determine the associations of physical capacity, psychological well-being, and ability to work in patients with post-COVID. For example, we will clarify whether patients with post-COVID symptoms and lower physical capacity also show lower values in neuropsychological health. Bottemanne et al. [3] previously revealed, that symptoms of depression after COVID-19 may be associated with persistent physical symptoms. Furthermore, we want to determine whether physical and/or psychological well-being are predictors for ability to work. Therefore, correlation as well as multivariate or logistic regression analysis and structural equation analysis will be considered in future.
- Magnavita N, Tripepi G, Di Prinzio RR. Symptoms in Health Care Workers during the COVID-19 Epidemic. A Cross-Sectional Survey. Int J. Env. Res. Public Health 2020, 17(14). Epub 2020/07/24. doi:10.3390/ijerph17145218.
- Delgado-Alonso, C.; Cuevas, C.; Oliver-Mas, S.; Díez-Cirarda, M.; Delgado-Álvarez, A.; Gil-Moreno, M.J.; Matías-Guiu, J.; Matias-Guiu, J.A. Fatigue and Cognitive Dysfunction Are Associated with Occupational Status in Post-COVID Syndrome. J. Env. Res. Public Health 2022, 19, 13368, doi:10.3390/ijerph192013368.
- Bottemanne, H., Gouraud, C., Hulot, J. S., Blanchard, A., Ranque, B., Lahlou-Laforêt, K., Limosin, F., Günther, S., Lebeaux, D., & Lemogne, C. Do Anxiety and Depression Predict Persistent Physical Symptoms After a Severe COVID-19 Episode? A Prospective Study. Frontiers in psychiatry 2021, 12, 757685. doi:10.3389/fpsyt.2021.757685.